# Analytical Scaling Laws for Radiofrequency-Based Pulse Compression in Ultrafast Electron Diffraction Beamlines

Paul Denham * and Pietro Musumeci

Department of Physics and Astronomy, University of California, Los Angeles, CA 90095, USA;
musumeci@physics.ucla.edu
* Correspondence: pdenham@physics.ucla.edu

**Abstract:** We present an envelope equation-based approach to obtain analytical scaling laws for the shortest pulse length achievable using radiofrequency (RF)-based bunch compression. The derived formulas elucidate the dependencies on the electron beam energy and beam charge and reveal how relativistic energies are strongly desirable to obtain bunches containing 1 million electrons with single-digit femtosecond pulse lengths. However, the non-linearities associated with the RF curvature and the beam propagation in drift spaces significantly limit the attainability of extreme compression ratios. Therefore, an additional higher frequency RF cavity is implemented, which linearizes the bunch compression, enabling the generation of ultrashort beams in the sub-femtosecond regime.

**Keywords:** longitudinal dynamics; beam manipulation; LINACs; ultrafast electron diffraction

## 1. Introduction

Ultrafast electron scattering requires the generation of very short electron bunches to capture the fastest physical processes [1,2]. Due to the repulsive effect of space-charge forces, one critical challenge in this field is related to packing as many electrons as possible in a short bunch [3]. In ultrafast electron diffraction (UED), pushing the electron energy to relativistic levels has helped in minimizing the space-charge effects, concurrently bringing other advantages such as longer penetration depths, reduced group velocity mismatch, and suppressed inelastic scattering background [4–7]. Over recent years, UED beamlines have seen continuous improvement in the achievable temporal resolution due to the introduction of techniques borrowed from accelerator physics based on the use of a time-dependent radiofrequency (RF) electric field to compress the electron bunch during its propagation in the beamline [8]. RF compression using 3 GHz resonant cavities has been applied to both non-relativistic and relativistic electron beamlines for UED [9–11], yielding bunch lengths down to the single-digit fs in the latter case.

While the discussion in this paper focuses on the electron bunch length, it is important to recognize that there are many additional factors, other than the temporal duration of the probe pulse, that contribute to the actual temporal resolution limit in a specific UED setup such as temporal jitter, group velocity mismatch, and laser pulse length. For example, to counteract the additional temporal jitter introduced by RF-based compression, naturally synchronized laser-generated higher-frequency waves have been used to impart an energy chirp on the beam in more complex coupling structures and drive compression dynamics [12,13].

In any case, though, to push the boundary of the UED technique, it is critical to understand the limits in beam compression and how the various beamline parameters such as charge, energy, cavity voltage, and frequency affect the shortest bunch duration achievable. The minimum bunch length at the sample results from a complex interplay between the details of the bunching dynamics and the longitudinal space-charge forces in the beam so that typically UED practitioners resolve to particle tracking simulation codes

to design the beamline and predict the beam dynamics. Generally, there is agreement with experimental results [14]. Still, particle simulations only deal with specific beamline setups, typically lack generality, and might not offer an immediate answer to how to improve compression in a given configuration.

It would be beneficial to have a unified formalism describing beam dynamics in RF-compression UED beamlines, covering both relativistic and non-relativistic cases while including the space-charge effects. To this end, we employ the longitudinal envelope equation formalism to highlight the interplay between longitudinal emittance and space-charge forces on pulse evolution. The single-particle dynamics presentation builds on previous works of Floettman and Zeitler [15,16] that pointed out the role of the non-linearities in the beam compression process. The collective effects are then considered in the approximation that the beam aspect ratio remains constant along the beamline, thus decoupling the longitudinal dynamics from the transverse beam size evolution. While this is a somewhat restrictive assumption, it is experimentally relevant (the beams in UED are usually focused transversely and longitudinally at the sample) because this approach yields an upper bound estimate for the minimum bunch length. In this case, space-charge forces are over-estimated for the situation in which the transverse spot size is kept large during the compression. Using the constant aspect ratio approximation, we can extend on the previous work and obtain analytical formulas for the minimum bunch length at the longitudinal waist that are valid in the presence of space charge. The expressions presented can then be used to guide the system optimization, compare parameter choices at different facilities, and evaluate mechanisms for further improving the bunch length.

The simple cartoon in Figure 1 illustrates the dynamics under study. Essentially, a finite electron beam propagates through an RF buncher cavity where electromagnetic fields oscillate with angular frequency $\omega = kc$ at the zero-crossing phase, where $k$ is the cavity wave number and $c$ is the speed of light. Ideally, the input bunch length satisfies $k\sigma_z << 1$, where $\sigma_z$ is the electron beam rms bunch length, so only a small phase window of the wave is sampled by the beam and the chirp imparted on the beam is predominantly linear. However, in our discussion, we keep the higher-order terms in the energy modulation expansion to elucidate their role in the final bunch length. In the propagation region after the buncher, due to the strong energy chirp, the tail of the beam begins to catch up, while the head of the beam slows down. Finally, at some location downstream of the buncher, ideally arranged to be the sample plane or the interaction point of the UED experiment, the minimum bunch length occurs when the phase space distribution is vertically aligned.

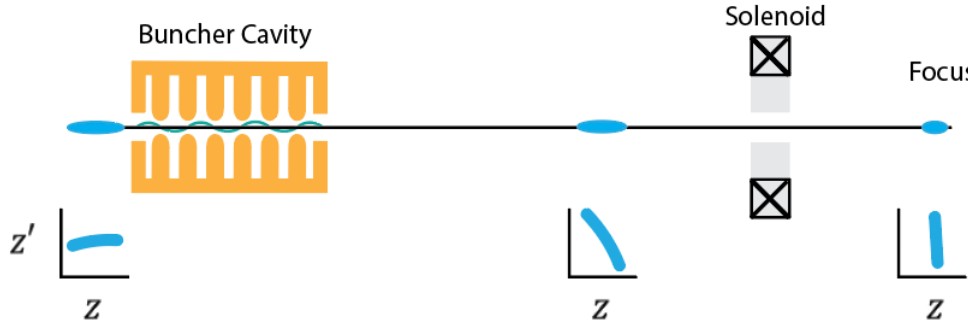

**Figure 1.** Illustration of RF ballistic bunching scheme. A velocity chirp is imparted on an electron using an RF cavity so that the tail of the beam has a higher energy than the head. During the following drift, the particles in the tail catch up with the particles in the head, resulting in strong longitudinal compression.

We strive to keep all the formulas in the paper as general as possible. For example, by not assuming the ratio of beam velocity to the speed of light, $\beta = 1$, so that formulas can be applied to different RF compression setups (non-relativistic, MeV UED beamlines, as well as higher-frequency compression schemes) once the parameters are scaled accordingly.

For this reason, we use two different example cases, loosely based on the UED beamlines at the UCLA Pegasus laboratory [10], to benchmark the agreement between the analytical framework and particle tracking simulations. The reference parameters used for this study are reported in Table 1.

**Table 1.** Simulation beam parameters.

| Parameter | High Energy | Low Energy |
|---|---|---|
| Focal length | 1.88 m | 1 m |
| Beam kinetic energy | 4.6 MeV | 150 keV |
| Norm. transverse emittance | 100 nm | 8.3 nm |
| RMS transverse beam size | 100 μm | 100 μm |
| Cavity Frequency | 2.856 GHz | 2.856 GHz |
| Relative energy spread | $10^{-5}$ | $10^{-5}$ |

The particle tracking software utilized to benchmark the analytical framework presented in this paper is General Particle Tracer (GPT) [14]. GPT is a full 3D particle tracking simulation software that enables a more complete study of 3D and non-linear effects of charged particle beam dynamics in electromagnetic fields and hosts a variety of visualization tools for windows-based computers. GPT also offers a selection of default fieldmaps of common beamline elements such as RF cavities and charged particle optics, which can be easily implemented to provide preliminary beamline models. Typically, as more realistic fieldmaps are generated, they can be implemented as custom elements in the simulation.

The paper is structured as follows. We first introduce the envelope equation formalism to describe the evolution of the bunch duration through the system [17]. We approximate the buncher as a thin lens, compute the main contribution to emittance growth and analyze the ballistic dynamics in the drift while neglecting space charge, thus essentially re-obtaining the results first presented in Floettmann [15], then in Zeitler et al. [16]. We then include the space-charge repulsion term in the envelope equation for a Gaussian beam using the approximation of a constant aspect ratio and arrive at an expression for the minimum achievable bunch length. We build on the formalism using it to describe different current profiles. Finally, in the light of the findings in this paper, we review the use of an additional higher-frequency RF cavity for compensation of non-linearities to reach sub-fs bunch lengths as originally proposed by Floettmann [15].

## 2. Longitudinal Envelope Equation

### 2.1. Envelope Equation

Among many possible choices for defining bunch duration (e.g., full width half maximum FWHM, or full width containing 50% of the charge FW50), in our discussion, we select the second-order moment of the longitudinal profile, or root mean square (RMS) bunch length, as the primary quantity to follow the evolution of in the RF compression beamline. The RMS is defined by

$$\sigma_z = \sqrt{\langle z^2 \rangle}, \tag{1}$$

where $z$ is the longitudinal particle coordinate relative to the center of the bunch. $\langle \ldots \rangle$ represents an expectation value over the beam distribution function. All the other definitions (i.e., FWHM and FW50) are simply proportional to the RMS bunch length using a proper order-of-unity pre-factor which depends on the beam distribution shape. The RMS beam size evolves along the beamline axial coordinate $s$ according to equation [18]

$$\sigma_z'' = \frac{\langle z z'' \rangle}{\sigma_z} + \frac{\epsilon_{z,z'}^2}{\sigma_z^3}, \tag{2}$$

where quantity $\epsilon_{z,z'}^2$ is the square of the RMS longitudinal trace space emittance, explicitly written in terms of the distribution moments as

$$\epsilon_{z,z'}^2 = \langle z^2 \rangle \langle z'^2 \rangle - \langle zz' \rangle^2, \tag{3}$$

and $z'$ is the relative velocity deviation from the average beam velocity. If the particle dynamics are linear in $(z, z')$ coordinates, then the trace space emittance is a conserved quantity [19].

Ideally, the longitudinal equation of motion does not depend on transverse coordinates. On the other hand, the force acting on the particles might have a transverse dependence (i.e., $z'' = \mathcal{F}(z, r, s)$) and in general the first term on the right side of Equation (2) becomes proportional to $\langle z\mathcal{F}(z, r, s) \rangle$. Heuristically, we separate the contributions to the longitudinal force into a term associated with the external RF fields and a term proportional to the beam current, which encodes the effect of the space-charge force, i.e., $\mathcal{F} = \mathcal{F}_{RF} + \mathcal{F}_{sc}$. After averaging over the transverse beam distribution, the latter depends on the horizontal and vertical RMS transverse sizes $\sigma_x = \sigma_y = \sigma_\perp$, effectively coupling the longitudinal and transverse envelope equations.

In most practical cases, we can model the RF buncher as a thin lens, and $\mathcal{F}_{RF}(z)$ provides an impulse force proportional to $\delta(s)$. In this case, the only effect of the RF buncher is to modify the initial conditions for the envelope equation through the application of a negative energy chirp to the phase space, which causes the beam to begin the bunching process (i.e., $\sigma'_{z0} < 0$). If the applied energy chirp is non-linear (which is the most common case), then the change in longitudinal emittance and the initial conditions must be evaluated at the exit of the buncher.

Meanwhile, the space-charge term $\langle z\mathcal{F}_{sc} \rangle (s, \sigma_z, \sigma_\perp)$ acts over the whole time of flight in the drift following the buncher. This term is related to the first-order Taylor expansion of the space-charge-induced longitudinal electric field near the center of the bunch. We discuss this in detail later in the paper, but we can anticipate some of the results derived below to orient this initial discussion. Typically, this term can be evaluated analytically for simple longitudinal distributions and written as

$$\frac{\langle z\mathcal{F}_{sc} \rangle (s, \sigma_z, \sigma_\perp)}{\sigma_z} = \frac{K_L}{\sigma_z^2}, \tag{4}$$

where $K_L \propto gNr_c/\beta^2\gamma^5$ is the longitudinal perveance proportional to the classical electron radius, $r_c$, the number of electrons in the bunch $N$, and $g$ is a geometry factor which depends on the beam aspect ratio.

In what follows, we assume a constant aspect ratio for the beam, a transverse RMS spot size that decreases along the beamline proportionally to the longitudinal RMS bunch length. This assumption simplifies the $s$ and $\sigma_\perp$ dependencies in $K_l$ and solves the longitudinal envelope equation independently from the transverse dynamics. The approximation works well even if the longitudinal and transverse spot sizes are not exactly proportional to each other along the beamline, but the aspect ratio remains within a factor of two of the initial value. Of course, this type of transverse focusing may not occur in all setups. In those cases, the transverse beam size remains large. Thus, the aspect ratio can increase by orders of magnitude during compression, and the approximation fails, but it does yield an overestimate of the waist size in the presence of space-charge effects. In addition, it is essential to note that for general beam distributions, the non-linearities in the bunch self-fields cause emittance growth during the propagation, violating the premises of our approach, which uses the envelope equation to treat the problem and assumes a constant longitudinal trace space emittance. Instead, one must self-consistently track all particles in the field generated by the charge distribution. Nevertheless, if one desires ultrashort bunch lengths, this situation should be avoided because space-charge effects should be relatively small.

### 2.2. Solution in a Drift

Let us start the discussion from the most straightforward and physically relevant case in which the propagation occurs in a drift and space-charge forces can be neglected, i.e., $\mathcal{F} = 0$. The shortest bunch length along the line can then be found by recasting Equation (2) as

$$\frac{1}{2}\frac{d}{ds}\left(\sigma_z'^2\right) = \frac{\epsilon_{z,z'}^2}{\sigma_z^3}\sigma_z',$$ (5)

which can be integrated exactly

$$\sigma_{zf}'^2 - \sigma_{z0}'^2 = \epsilon_{z,z'}^2\left(\frac{1}{\sigma_{z0}^2} - \frac{1}{\sigma_{zf}^2}\right).$$ (6)

The waist position is a local minimum for $\sigma_z$; thus, we can set $\sigma_{zf}' = 0$. Right after the buncher, we can write $\sigma_{z0}' = \frac{\langle z_0 z_0'\rangle}{\sigma_{z0}} - \frac{\sigma_{z0}}{f}$ where the first term accounts for any incoming correlations in the phase space and the second term accounts for the linear chirp imparted by the buncher cavity. After substituting into Equation (6), the waist size can be written as

$$\sigma_{zf} = \frac{1}{\sqrt{\frac{1}{\epsilon_{z,z'}^2}\left(\frac{\langle z_0 z_0'\rangle}{\sigma_{z0}} - \frac{\sigma_{z0}}{f}\right)^2 + \frac{1}{\sigma_{z0}^2}}} \approx \frac{f\epsilon_{z,z'}}{\sigma_{z0}}\left|1 - \frac{f\langle z_0 z_0'\rangle}{\sigma_{z0}^2}\right|^{-1},$$ (7)

where we neglect the term $\frac{1}{\sigma_{z0}^2}$ which for large compression factors is always much smaller than $\frac{1}{\sigma_z^2}$.

By inspecting the factor accounting for initial correlations, we can see that an initial negative chirp (i.e., $\langle z_0 z_0'\rangle < 0$) effectively shortens the focal length and final bunch length in the system. On the other hand, in most cases, an RF buncher is added to the system to achieve strong compression and $\frac{f\langle z_0 z_0'\rangle}{\sigma_{z0}^2} \ll 1$, i.e., incoming correlations are small compared to the linear correlation imparted by the RF fields. In this case, the final bunch length at the waist is given by $\frac{f\epsilon_{z,z'}}{\sigma_{z0}}$ and is simply proportional to the focal length times the energy spread, assuming that thermal contributions dominate the longitudinal emittance. However, we see that the non-linear correlations imparted by the buncher significantly distort the trace space and dominate the emittance in the final drift. Ultimately, beams with smaller longitudinal emittance enable reaching shorter bunch lengths. It also follows that one can achieve proportionally shorter final bunch durations by decreasing the focal length $f$ of the RF buncher.

### 2.3. Single Particle Dynamics and Non-Linear Phase-Space Correlations in the RF Buncher

Let us now look more closely at the details of the energy chirp imparted by the RF buncher on the beam distribution. The main assumption here is that the cavity fields act on the electrons by adding an energy kick with sinusoidal dependence on the initial longitudinal particle position $z_0$ as

$$\Delta\gamma = -\alpha\sin(kz_0),$$ (8)

where $\alpha = eV_0/mc^2$ and $eV_0$ is the cavity voltage or the maximum energy gain seen by an ideally phased particle, and $k = k_0/\beta$ is the RF angular wave number divided by the normalized longitudinal velocity. The phase of the cavity is tuned so that the center of the bunch experiences no net energy gain and particles at the tail gain energy, while particles at the head of the bunch lose energy. There are two distinct sources of non-linearities in the trace space dynamics resulting from the applied energy change to the particles. First, for finite duration input bunches, the curvature of the RF wave causes significant non-

linear effects in the trace space. In addition, the relativistic relation between normalized velocity and beam energy $\beta = \sqrt{1 - 1/\gamma^2}$ adds an important degree of non-linearity to the transport as pointed out in Zeitler et al. [16]. Following the discussion therein, Taylor-expanding the relative velocity deviation $\frac{\Delta\beta}{\beta}$ in terms of the energy deviation, we can write

$$\frac{\Delta\beta}{\beta} = \sum_m \eta_m \Delta\gamma^m, \tag{9}$$

where $\eta_m$ are proportional to the mth derivatives $\frac{d^m}{d\gamma^m}\beta$ and in particular

$$\eta_1 = \frac{1}{\beta^2\gamma^3}, \tag{10a}$$

$$\eta_2 = \frac{2 - 3\gamma^2}{2\gamma^6\beta^4}, \tag{10b}$$

$$\eta_3 = \frac{2 - 5\gamma^2 + 4\gamma^4}{2\gamma^9\beta^6}, \tag{10c}$$

where $\gamma$ and $\beta$ are the mean values of the normalized energy and velocity distributions, respectively. Coefficients $\eta_m$ scale as $\gamma^{-(m+2)}$ so that at high relativistic energies, the higher-order non-linear terms in the transport can be neglected. Considering the lowest-order dynamics, we can simply replace $\sin(kz_0)$ with $kz_0 - \frac{(kz_0)^3}{6}$, and truncate the series to obtain

$$\frac{\Delta\beta}{\beta} \approx -\eta_1\alpha(kz_0) + \eta_2\alpha^2(kz_0)^2 + \left(\frac{\eta_1\alpha}{6} - \eta_3\alpha^3\right)(kz_0)^3. \tag{11}$$

We verify this expression at high energy (4.6 MeV) and low energy (150 keV) by considering a particle tracking simulation of the buncher configuration listed in Table 1 with an initial bunch length of 195 µm and 1.87 mm, respectively. The buncher is modeled by a 2.856 GHz standing wave cylindrically symmetric TM010 cavity, supporting the first transverse magnetic standing wave mode with an axial electric field component for particle acceleration, with an amplitude adjusted to reach a longitudinal focus 1.88 m and 1 m downstream, respectively, for the high- and low-energy cases. Since the RF cavity length is 0.05 m, it is reasonable to approximate it as a thin lens. The longitudinal phase spaces from GPT at the exit of the buncher are shown in Figure 2a,c for the high- and low-energy cases, respectively, with subtracted linear correlations. The quality of the agreement between GPT and our analytical framework can be assessed by comparing the distributions with the lines corresponding to Equation (11) which are also shown. The parameters chosen for these examples highlight the different possibilities for the dominant non-linearity in the system. In the high-energy case, the relativistic effects are responsible for the parabolic shape seen in the simulation, while in the low-energy case, the injected bunch length is longer, and the third-order non-linearity associated with the sinusoidal RF fields is the main effect in the beam distribution shape.

The convenience of working in the trace space is the linearity of the dynamics in the drift which fully preserves the trace space area. Explicitly, in the drift after the buncher, the longitudinal particle position can be written as

$$z = z_0 + s\frac{\Delta\beta}{\beta} = z_0 + s\sum_{n=1}^{\infty} \eta_n \Delta\gamma^n. \tag{12}$$

The initial coordinate $z_0$ is expressed in terms of the induced energy modulation $\Delta\gamma$ by inverting Equation (8). Then, $\Delta\gamma$ is Taylor expanded in terms of $\Delta\beta/\beta$. Substituting into Equation (12), keeping only terms up to the third order, we can write

$$z \approx \left(s - \frac{1}{\eta_1\alpha k}\right)\frac{\Delta\beta}{\beta} + \frac{\eta_2}{\eta_1^3\alpha k}\frac{\Delta\beta^2}{\beta^2} - \frac{\left(\frac{\eta_1\alpha}{6} - (\eta_3 - 2\eta_2^2/\eta_1)\alpha^3\right)}{\eta_1^4\alpha^4 k}\frac{\Delta\beta^3}{\beta^3}. \tag{13}$$

The longitudinal waist occurs where the linear chirp is cancelled at distance $s = \frac{1}{\eta_1 \alpha k}$ along the beamline, allowing us definition of the buncher longitudinal focal length

$$f = \frac{1}{\eta_1 \alpha k} = \frac{m_0 c^2 \gamma^3 \beta^2}{e V_0 k}, \tag{14}$$

which indicates that very high voltage cavities are needed to obtain short focal lengths for relativistic electrons. It is also useful to note the k-dependence of this expression which favors the use of very high frequencies for this application.

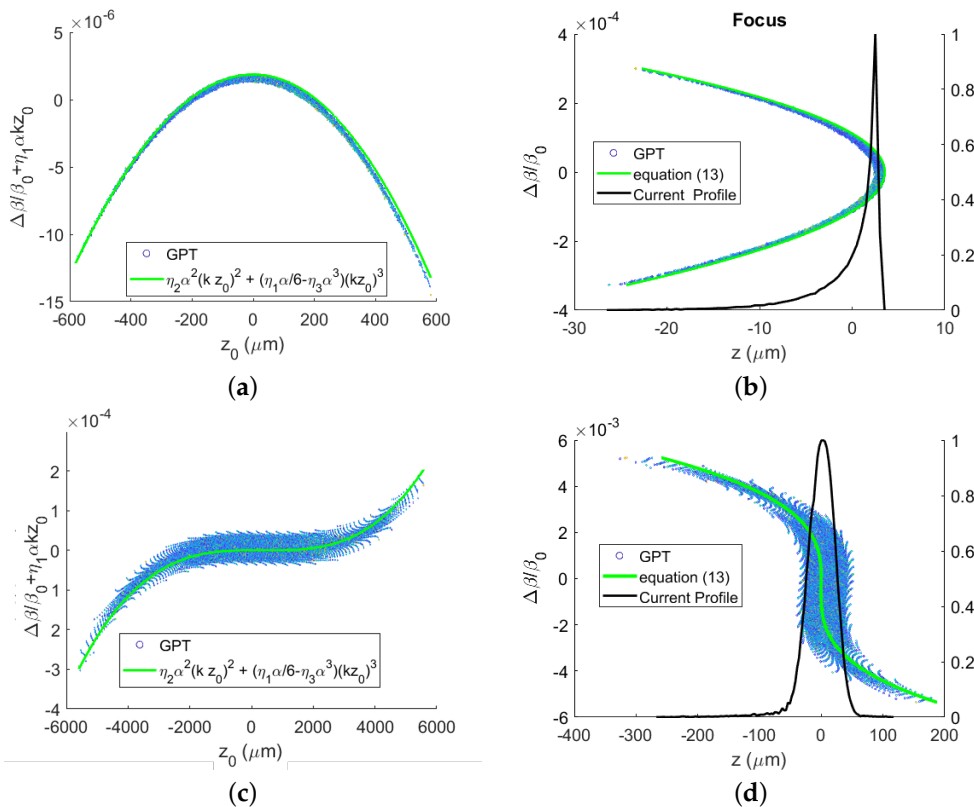

**Figure 2.** (**Left**) Trace spaces of the beam at the exit of the prebuncher after the linear chirp is subtracted from the distribution for high-energy (**a**) and low-energy (**c**) cases compared with the analytical predictions from Equation (11) (green curves). (**Right**) Longitudinal trace spaces at the temporal waist for the high-energy (**b**) and low-energy (**d**) cases compared with the predictions from Equation (13) (green curves). The current profiles at the focus are also shown in black.

At the focal plane, the residual correlation is quadratic or cubic in $\Delta \beta$ depending on the relative importance of the non-linearity in the drift propagation concerning the RF curvature. As discussed above, lower beam energies and longer input bunches tend to show higher third-order non-linearities, while relativistic energies typically have dominant second-order contributions. Predictions from Equation (13) can be again verified by comparison to the phase spaces at the temporal waist plane from the same GPT simulation, as shown in Figure 2b,d.

### 2.4. Emittance Growth Mechanisms and the Relationship between Different Longitudinal Phase Space Definitions

#### 2.4.1. $(z, z')$ Trace Space Emittance

Since the drift dynamics in the trace space are entirely linear, and the emittance growth is all accrued in the buncher, the envelope equation formalism is a convenient choice to follow the RMS bunch length evolution. To evaluate the RMS emittance growth induced by the RF compressor, we start from an initial longitudinal phase space with RMS emittance

$\epsilon_{z_0, z_0'}$. After the energy chirp is applied, the single-particle velocity variation maps to $z_0' \to z_0' + \Delta\beta/\beta$, where $\Delta\beta/\beta$ represents the velocity variation imparted by the buncher, which is correlated with particle position. In the thin lens approximation, the particles do not change position as the beam proceeds through the cavity.

The moments of the new distribution can be calculated and the relation between initial and final emittance after the buncher written as

$$\epsilon_{zz'}^2 = \epsilon_{z_0 z_0'}^2 + \epsilon_{RF}^2$$

$$= \epsilon_{z_0 z_0'}^2 + \langle z_0^2 \rangle \langle \left( \frac{\Delta\beta}{\beta} \right)^2 \rangle - \langle z_0 \left( \frac{\Delta\beta}{\beta} \right) \rangle^2.$$

We note that each respective expectation value needed to calculate the emittance, with the exception of $\langle z_0^2 \rangle = \sigma_{z0}^2$, requires integrating $\sin^m(kz)$ or $z \sin^m(kz)$ over the beam distribution. Assuming an initial Gaussian current profile, these integrals have closed-form expressions up to arbitrary order of m, but keeping the leading contributions to the emittance growth, we obtain

$$\epsilon_{RF}^2 \approx \sigma_{z0}^2 \left[ 2\eta_2^2 \alpha^4 k^4 \sigma_{z0}^4 + \frac{1}{6} \left( \eta_1 \alpha - 6\eta_3 \alpha^3 \right)^2 k^6 \sigma_{z0}^6 \right]. \tag{15}$$

In most cases, this expression is much larger than the initial longitudinal emittance because the buncher is fundamentally inducing a large velocity spread (with correspondingly significant non-linear contributions) to achieve strong compression. As long as the space-charge forces are negligible in the drift, $\epsilon_{RF}$ is also equal to the final emittance and can be used to calculate the shortest bunch length achievable at the waist using Equation (7). The emittance growth in the buncher and its preservation in the drift are shown in Figure 3, where the trace space RMS emittance evolution calculated from GPT is plotted along the beamline. The black dotted line shows Equation (15), which provides a good approximation for the final emittance after a thin lens buncher cavity located at the origin.

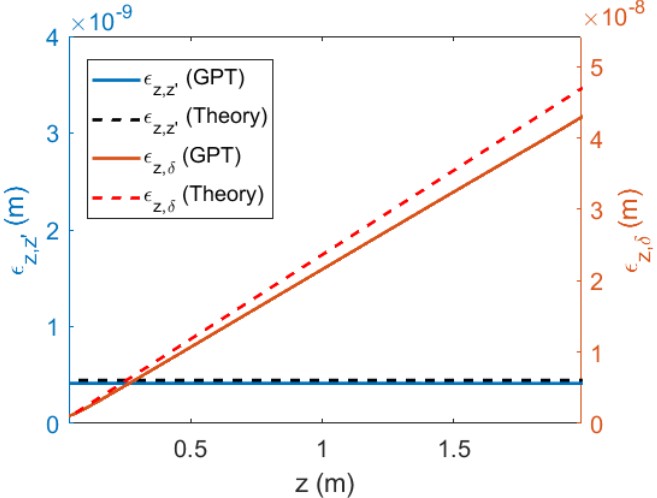

**Figure 3.** Comparison between analytical estimates (dashed lines) and GPT simulations (solid lines) for an initial bunch length of 0.65 ps. The trace space emittance after the buncher is shown in blue, and emittance growth in $(z, \delta)$ phase space is shown in orange.

### 2.4.2. $(z, \delta)$ Phase Space

It is important to note at this point that if instead we utilized more common choices of defining the longitudinal trace space in terms of the relative energy spread $\delta = \frac{\Delta\gamma}{\gamma}$ or momentum spread $\Delta p_z / p_z$, the drift dynamics would become highly non-linear, especially for mildly relativistic particles.

Initially, the trace space emittance $\epsilon_{z,z'}$ is related to the $(z,\delta)$ emittance by the following relationship:

$$\epsilon_{z_0,\delta} = \gamma^2 \beta^2 \epsilon_{z_0,z_0'}. \tag{16}$$

This relationship holds because $z' \approx \eta_1 \Delta\gamma$ and explains the order of magnitude difference in absolute emittance values in Figure 3. It is then fairly common to see in the literature the envelope equation written in terms of the $(z,\delta)$ emittance:

$$\sigma_z'' = \frac{\langle zz'' \rangle}{\sigma_z} + \frac{\epsilon_{z,\delta}^2}{\beta^2 \gamma^2 \sigma_z^3}. \tag{17}$$

Nevertheless, in a drift, the particle positions evolve according to Equation (12) where the transport is inherently non-linear, especially for mildly relativistic particles, causing emittance growth and limiting the usefulness of the envelope equation approach. Ultimately, for large enough initial energy spreads, the higher-order terms proportional to $\eta_m$ lead to significant $(z,\delta)$ emittance growth, which can be estimated using the same techniques as in the previous subsection:

$$\epsilon_{z\delta}^2 = \epsilon_{z_0,\delta_0}^2 + 2s^2 \eta_2^2 \alpha^6 k^6 \sigma_z^6, \tag{18}$$

where $\epsilon_{z_0,\delta_0}$ is the emittance at the beginning of the drift. This expression predicts a nearly linear growth with propagation distance for a small initial emittance. This is shown in Figure 3 where the $(z,\delta)$ phase space evolution from GPT is compared with Equation (18) with the inclusion of the initial emittance as well.

The seemingly counterintuitive behavior of the longitudinal emittance (growing linearly in the drift) is the main reason we adopt the $(z,z')$ trace space emittance in calculating the final bunch length when using the envelope equation formalism. Finally, for completeness, we observe that if the un-normalized momentum $(z, \Delta p_z / p_z)$ was used as a trace space variable, all expressions could be simply modified substituting $z' = \frac{1}{\gamma_0^2} \Delta p_z / p_z$. Nevertheless, due to the relativistic non-linear relation between momentum and velocity, even in this case, we would have significant emittance growth in drift propagation.

*2.5. Bunch Length Limit in Absence of Space-Charge Effects*

This formalism clarifies how the minimum achievable bunch duration depends on the main beamline parameters (when space-charge effects can be neglected). Inserting the trace space longitudinal emittance estimate (Equation (18)) into the envelope-equation solution (Equation (7)), we obtain an expression written as the quadrature sum of the different contributions to the final emittance between (i) the initial uncorrelated relative energy spread $\sigma_\delta$, (ii) the non-linearities introduced by the relativistic correction to the transport or (iii) the RF-induced emittance.

In order to facilitate comparison between beamlines of different energies, we can rewrite the terms as a function of the buncher's focal length, which is a practically helpful parameter related to standard requirements on dimensions of the sample chamber, pumping geometry, and transverse optics. Assuming that one term in the sum is much larger than the others, we can synthesize this result as

$$\sigma_{zf} \approx \max \begin{cases} f\sigma_{z0'} = \frac{f}{\beta^2 \gamma^2} \sigma_\delta \\[2mm] \sqrt{2} \frac{|\eta_2|}{\eta_1} \alpha k \sigma_{z0}^2 \approx \frac{3\sqrt{2}\gamma^2}{2} \frac{\sigma_{z0}^2}{f} \\[2mm] \frac{1}{\sqrt{6}} k^2 \sigma_{z0}^3. \end{cases} \tag{19}$$

We note that the total quadrature sum of these three expressions should be used in those cases where two or more contributing terms have a similar magnitude.

In the first case, non-linearities in the transport can be neglected, and the final bunch length is simply proportional to the initial relative velocity spread $\sigma'_{z0}$. For the same uncorrelated relative energy spread $\sigma_\delta$, a relativistic energy system has a clear advantage to achieve ultrashort bunch lengths due to the inverse square $\gamma$ dependence in the formula. In reality, as the energy increases, the non-linearities due to the relativistic dynamics in the drift would likely become the dominant contribution to the final emittance and bunch length. In this regime, after approximating $\eta_2/\eta_1^2 \approx 3/2\gamma^2$ and using the definition of the focal length (Equation (14)), the final bunch duration scales as the square of the beam energy. For longer initial bunch lengths, the curvature of the RF dominates the shape of the final phase space and sets the limit for the shortest bunch duration achievable. This contribution is essential for both relativistic and non-relativistic energies, and it strongly favors the use of lower RF frequencies. In all THz compression experiments carried out so far, this term has been the principal limit to the final bunch length [12,20].

The analytical formulas summarized in Equation (19) were found in excellent agreement with GPT in both the relativistic and non-relativistic regimes. The results are shown in Figure 4, where (a) is corresponds to the 4.6 MeV and (b) the 150 keV cases, respectively. In the low-energy case, the cubic non-linearity from the RF curvature dominates due to the longer initial bunch length.

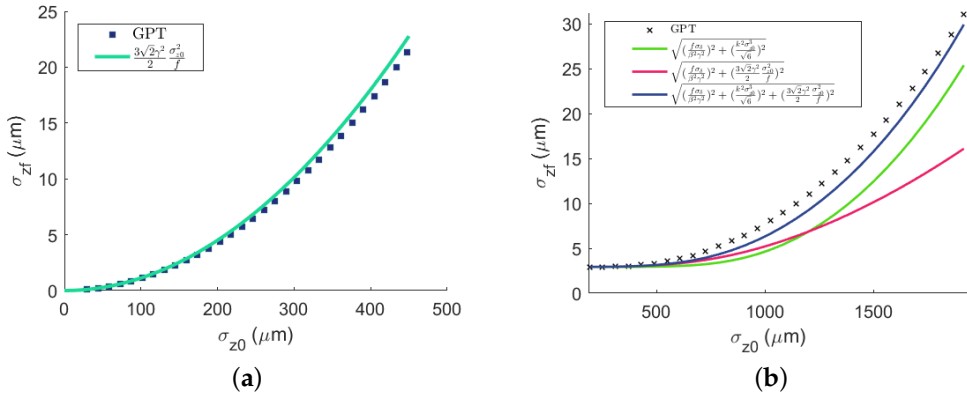

**Figure 4.** Final bunch length as a function of the initial bunch length for the (**a**) high-energy and (**b**) low-energy cases. The analytical curves are also shown and are found to be in very good agreement with the simulations.

In the absence of space charge, the shortest bunch durations are obtained by minimizing the emittance growth in the temporal lens, which can be achieved by using shorter input beams. At the same time, as we see further, decreasing the initial bunch length also increases the initial peak current so that the space-charge effects in the longitudinal envelope equation can no longer be neglected. As a result, space-charge effects begin to take over at a certain point and prevent further bunch compression. Thus, there must be an optimum initial bunch length to inject, which exactly balances space charge and RF emittance growth.

## 3. Space-Charge Limits to Compression

### 3.1. An Example of Geometry Factor Calculation: Gaussian Distribution Case

In order to add the effect of space charge on the bunch length evolution into the envelope equation formalism, we need to compute a reasonable representation for the $\langle z \mathcal{F}_{sc} \rangle (z, \sigma_z, \sigma_\perp)$ term.

For simplicity, we derive the self-field of an azimuthally symmetric 3D Gaussian beam assuming that the distribution function remains a Gaussian profile throughout evolution. Admittedly, this approximation is especially poor at the waist where the linear chirp is removed, and what remains is a non-linear distribution in trace space with a characteristic current spike and a temporal profile strongly asymmetric and far from a regular Gaussian.



In this situation, the emittance growth caused by the higher-order terms in the electric field profile also makes the envelope equation approach less helpful in describing the bunch evolution, and we would need to revert back to self-consistent particle tracking simulations.

Nonetheless, the simple expression for the force obtained below can at the very least be used to estimate the evolution of the bunch length and the point in the system where space-charge effects become dominant in the dynamics. To validate all of the estimates made in what follows, we compare the results with simulations of the bunching process utilizing GPT's spacecharge3Dmesh algorithm for the nominal high-energy beam parameters shown in Table 1 and a beam charge of 16 fC ($10^5$ electrons).

The charge density in the beam rest-frame can be written as

$$\rho = \frac{Q \exp\left(-\frac{r^2}{2\sigma_r^2} - \frac{z^2}{2\sigma_z^2}\right)}{(2\pi)^{3/2}\sigma_r^2\sigma_z}. \tag{20}$$

In order to obtain the electric field components, we Fourier transform the charge density and write the potential as

$$\phi(x,y,z) = \iiint \frac{\tilde{\rho}(k_x,k_y,k_z)}{\epsilon_0(k_x^2 + k_y^2 + k_z^2)} \exp(i\boldsymbol{k}\cdot\boldsymbol{r})\frac{d^3k}{(2\pi)^3}, \tag{21}$$

where $\tilde{\rho}$ is the Fourier transform of the charge density which also has a Gaussian shape.

We can then expand the complex exponential as a Taylor series in $(\boldsymbol{k}\cdot\boldsymbol{r})$. After dropping the constant term, which is immaterial for the field profile, we also note that all the odd terms vanish by the symmetry of $\tilde{\rho}$. The second-order term yields the uncorrelated linear electric field components. They can be written as

$$\boldsymbol{E}^{(1)} = \frac{Q/2\sigma_z}{(2\pi)^{3/2}\epsilon_0\sigma_\perp^2}u(A)r\hat{\boldsymbol{r}} + \frac{Q}{(2\pi)^{3/2}\epsilon_0\sigma_z^3}v(A)z\hat{\boldsymbol{z}}, \tag{22}$$

where $A = \sigma_\perp/\sigma_z$ is the beam aspect ratio (in its rest frame). The predicted field gradients are found in excellent agreement with core field gradients extracted from GPT simulation as shown in the example in Figure 5a. The geometry factors (plotted for reference in Figure 5b are given by

$$u(A) = \frac{\xi(A) - (1 - \xi(A)^2)\coth^{-1}\left(\frac{1}{\xi(A)}\right)}{\xi(A)^3}, \tag{23}$$

$$v(A) = \frac{\coth^{-1}\left(\frac{1}{\xi(A)}\right) - \xi(A)}{\xi(A)^3}, \tag{24}$$

where $\xi(A) = \sqrt{1 - A^2}$.

The generalized force term in the envelope equation can be calculated using $z'' = \gamma'/\gamma^3\beta^2 = eE_z/\gamma^3\beta^2mc^2$ and then taking the average over the beam distribution of $\langle zE_z\rangle$. Before doing so, we express the longitudinal electric field just calculated in the beam rest frame in terms of laboratory frame quantities. This can be achieved by rescaling the longitudinal coordinate and RMS moments by $\gamma$, i.e, $E_z(z,\sigma_z,\sigma_\perp) \to E_z(\gamma z, \gamma\sigma_z, \sigma_\perp)$. Finally, the space-charge term in the envelope equation is given by

$$\frac{\langle z\mathcal{F}_{sc}(z,\sigma_z,\sigma_\perp)\rangle}{\sigma_z} = \frac{g(A)Nr_c}{\beta^2\gamma^5\sigma_z^2} = \frac{K_L}{\sigma_z^2}, \tag{25}$$

where $g(A) = \frac{1}{2\sqrt{\pi}}v(A)$ is an order of unity factor which takes into account the geometry effects, $r_c$ is the classical electron radius and $N$ the number of electrons in the bunch; thus, we define the longitudinal perveance $K_L$ as anticipated in the first section. In Figure 5a, we compare the result of averaging the longitudinal electric field over the beam distribution

with the on-axis linear component, showing how averaging reduces the slope of the $z$, $E_z$ correlation by the constant term $\sqrt{2}/4$.

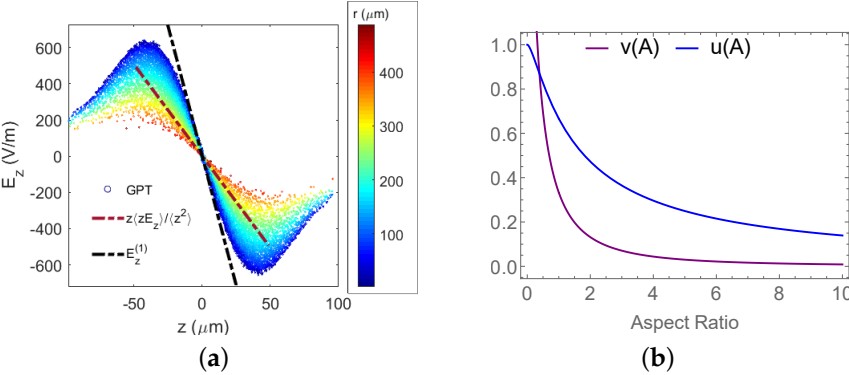

**Figure 5.** (**a**) Longitudinal field of the bunch compared with the linear field component of the 3D Gaussian field and with the beam-averaged generalized force term that appears in the envelope equation. The total charge in the bunch in this simulation is $Q = 10^5 e$. (**b**) Geometry factors for the transverse (blue) and longitudinal (purple) field plotted as a function of the rest frame aspect ratio.

In general, the dependence of the geometry factor $g(A)$ on the beam aspect ratio requires self-consistently solving the transverse and longitudinal envelope equations as a coupled system, which typically can be achieved through numerical integration.

An attractive simplification occurs if we use the approximation that the aspect ratio remains constant (or nearly constant) throughout the beamline, which corresponds to the time when the beam is simultaneously transversely focusing and longitudinally compressing. In this case, the system is decoupled, and we can then utilize $g(A)$ to integrate the longitudinal envelope equation analytically. This approximation turns out to be quite acceptable, as in many UED setups it is required to have a small transverse spot at the sample in the same plane of the temporal waist.

In Figure 6, we plot the evolution of the bunch length and transverse spot along the beamline in the case where optics are arranged so that the aspect ratio during compression is kept close to unity. For this case, we also use GPT simulations to numerically compute the exact expression for the space charge term in the envelope equation at any given location along the beamline and find good agreement with the analytical expression in Equation (25) with $A = 1$.

Importantly, when transverse focusing is not applied (or weak) as the beam reaches its minimum bunch length, the aspect ratio increases to values larger than unity along the beamline. Indeed, Figure 5b suggests that our approximation would lead to overestimating the space-charge force.

*3.2. Effect of the Longitudinal Space-Charge Force on the Minimum Bunch Length*

Using these results, the longitudinal envelope equation in presence of the space charge can be written as

$$\sigma_z'' = \frac{K_L}{\sigma_z^2} + \frac{\epsilon_{z,z'}^2}{\sigma_z^3},\tag{26}$$

and following the same steps that led to Equation (7), we can directly integrate to calculate the bunch length at the waist, obtaining

$$\sigma_{zf} = \frac{\epsilon_{z,z'}^2}{\sqrt{\frac{\sigma_{z0}^2 \epsilon_{z,z'}^2}{f^2} + K_L^2} - K_L},\tag{27}$$

where we once again assume that the initial bunch length is much larger than the final one.

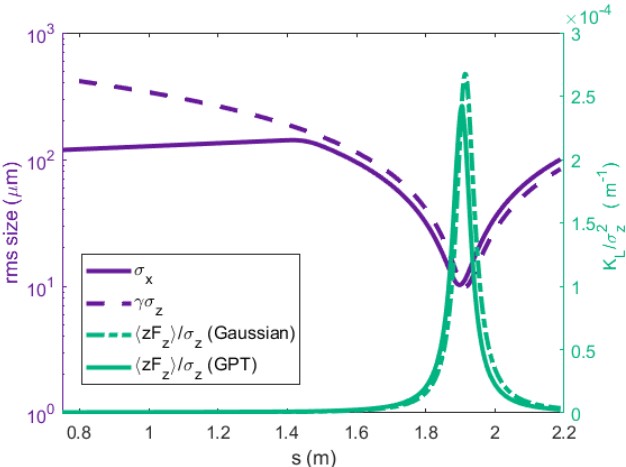

**Figure 6.** The evolution of longitudinal and transverse beam sizes are shown in purple. The space charge envelope equation term is shown in green, and its approximation is shown as a dotted line with aspect ratio fixed, i.e., A = 1.

In the limit that $K_L$ becomes negligible, Equation (27) yields back the zero space-charge solution discussed in the previous section. Conversely, if space charge dominates the bunch length evolution, one can expand the formula for large $K_L$ to obtain

$$\sigma_{zf} = \frac{2f^2 K_L}{\sigma_{z0}^2},$$ (28)

which is linear in the bunch charge but most importantly decreases as $\sigma_{z0}^2$, driving the initial bunch length towards larger values. We note that if by some clever scheme (for example, by pre-compensating using an X-band RF cavity as discussed in the last section of the paper) the RF emittance growth in the buncher is eliminated, a longer initial bunch length could be used, reducing space-charge effects and potentially allowing for reaching very short bunch lengths.

The optimal initial conditions are to compromise between the tendency to minimize the buncher's RF emittance growth and lowering the initial peak current. An estimate for the ideal initial bunch length can be obtained by equating the asymptotic dependencies of the waist size in the space-charge dominated regime and the emittance dominated regime, respectively.

For example, in the case where the RF emittance is dominated by the quadratic non linearity, we can set Equation (28) equal to the second expression in Equation (19), and the optimum initial bunch size is given by

$$\sigma_{z0}^4 \cong \frac{\sqrt{2}K_L}{\eta_1 |\eta_2| \alpha^3 k^3}.$$ (29)

This result can be substituted into Equation (27), yielding for the minimum bunch duration at the temporal waist:

$$\sigma_{zf} \cong 2.72 \sqrt{\frac{K_L |\eta_2|}{\eta_1^3 \alpha k}}.$$ (30)

In the high energy limit ($\gamma_0 \gg 1$),

$$\sigma_{zf} \sim \sqrt{\frac{3gNf r_c}{\gamma_0^3}}.$$ (31)

In Figure 7a, we show the final waist as a function of the injected bunch length obtained from GPT, and overlay Equation (27) for the case when the total charge is $Q = 10^5 e = 16$ fC. The dashed red line is the estimate obtained from Equation (30), which predicts an RMS size of 0.8 μm, underestimating by 20% the GPT prediction at 1 μm. For large enough initial bunch length, space-charge effects are negligible, and the emittance growth in the buncher dominates the waist size. When the initial bunch length is smaller than the optimum, i.e., in the space-charge dominated regime, the minimum bunch length increases as predicted by Equation (28). In Figure 7b, the phase space at the waist for the optimal injection case is shown. The quadratic correlation in the phase space is still visible, but an imprint of the space-charge field also appears.

The slight discrepancy between the simulation and the analytical prediction is due to the unaccounted space-charge-induced emittance growth. As we shorten the initial bunch length, the non-linearities in the space charge field are responsible for significant emittance growth, thus disrupting the applicability of the analytical result, which relies on the assumption of constant emittance in the drift. This is elucidated in Figure 7c,d, where we show the trace space emittance evolution for a few different compression cases and the final emittance as a function of input bunch length, respectively. In (c), it can be seen that the longitudinal emittance after the buncher is changing in the drift due to non-linear space-charge forces. The increase is modest when the input size is sufficiently large and more pronounced for smaller initial bunch lengths. In (d), the longitudinal emittances calculated from Equation (15) are compared with the final emittances at the waist from GPT. The emittance growth induced by the space charge is proxied by subtracting the RF growth from the emittance at the focus position in quadrature. Therein, it can be seen that the optimum input bunch length (72 μm, in this case) occurs at the onset of the space-charge-induced emittance growth. Eventually, when the initial bunch length becomes too short, the emittance growth is dominated by the space charge, so the GPT simulation results are not identical to the analytical predictions.

In Figure 8a, the comparison of Equation (30) with GPT simulations performed while setting the initial bunch length to Equation (29) as the charge is varied from $10^5 e \rightarrow 10^6 e$ shows good agreement when using a scaled longitudinal perveance to take into account the emittance growth from non-linear space-charge forces. In Figure 8b, we utilize Equation (30) to visualize the dependence on energy and charge when the focal length of the bunching system is set to 1.88 m. Due to the energy dependence intrinsic in the focal length, as we increase the beam energy, keeping the focal length constant becomes a technological feat involving considerations of the breakdown limit in RF cavities and available power sources at higher frequencies.

We note that the optimal initial bunch length identified above is the RMS spot size at the entrance of the buncher. Depending on the gun technology employed to generate the electron beam, there might be different ways to tune this quantity (i.e., changing the laser pulse length, operating at different phase or gradient in an RF gun, controlling the transverse dynamics), which might have an additional effect on the actual beam emittance or phase space correlations.

To move beyond these limits, we would need to reduce the space-charge effects (working to either minimize the geometry factor $g$ in the longitudinal perveance or utilize a charge distribution with more linear self-fields) or minimize the emittance growth in the buncher, for example, pre-compensating for the non-linearities in the input phase space. In principle, both options are feasible and are the subjects of the last two sections of this paper.

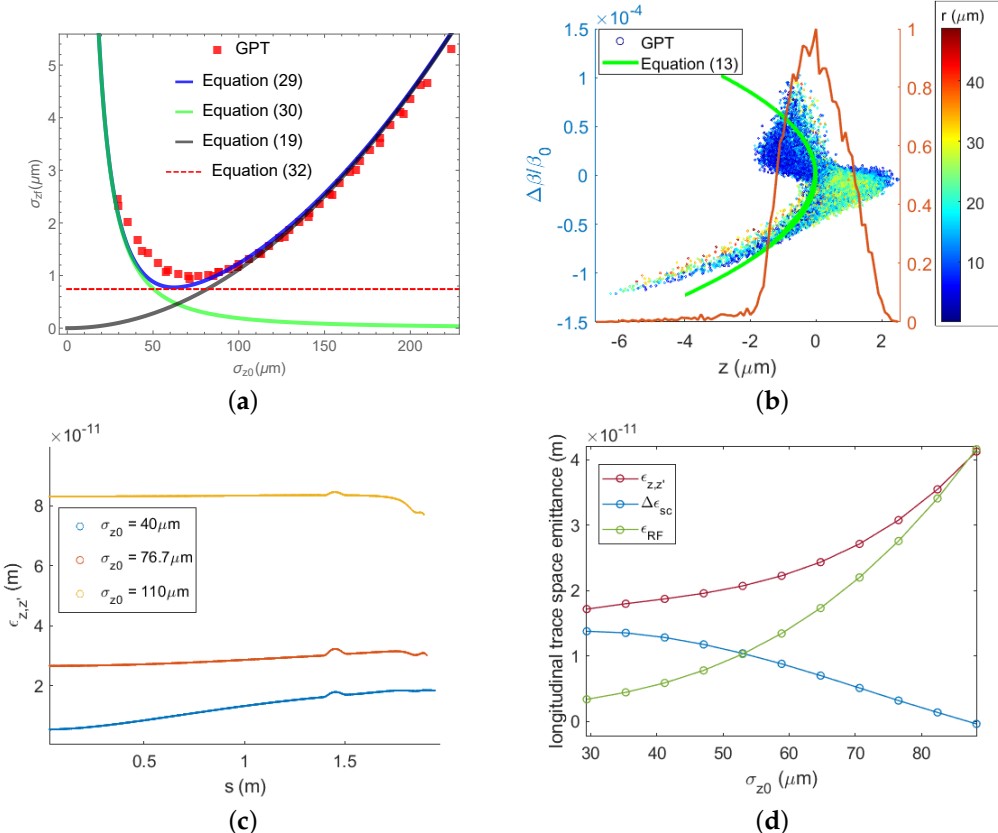

**Figure 7.** (**a**) Final waist size plotted versus initial bunch length compared with GPT. The green and black lines show the asymptotic behavior in the regimes where space charge and RF emittance growth dominate the dynamics. The blue line is the analytical expression that considers all effects discussed in this paper (**b**) The optimum phase space found from the scan in (**a**) for an input bunch length of 72 µm. The RMS bunch length is 1 µm. Particles are color-coded as a function of their radial coordinate (blue corresponds to on-axis). (**c**) Evolution of the emittance along the line. The growth observed in GPT is due to the non-linearities of the space-charge field. (**d**) Final emittance in GPT as a function of initial bunch length. The relative importance of the space-charge contribution to the emittance can be inferred by subtracting the expected emittance growth due to the buncher dynamics non-linearities. The cross-over point (where space charge becomes the dominant effect) can be used to estimate the optimal injection condition and hence the minimum final bunch length achievable for a given setup.

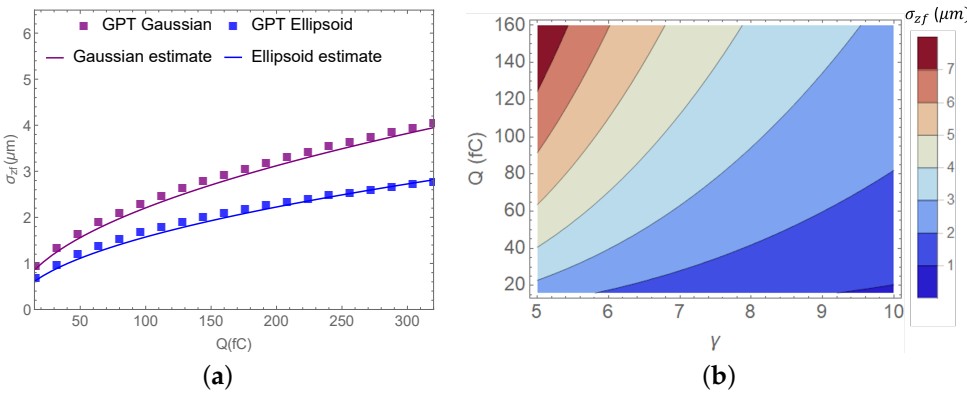

**Figure 8.** (**a**) Minimum bunch length analytical estimates compared with GPT simulation as the charge in the 4.6 MeV energy beam is varied for Gaussian (purple) and uniformly filled ellipsoidal (blue) beam distributions. (**b**) Minimum bunch length versus $\gamma$ and $N$ for the Gaussian distribution assuming a constant focal length $f = 1.88$ m.

We should also observe that the presented solution of the envelope equation by direct integration does not yield the position of the longitudinal waist along the beamline, which is an essential experimental parameter in the case where space-charge forces significantly contribute to the dynamics. The space charge pressure pushes the waist position downstream of the zero charge case, with a shift increasing for more significant beam charges, but that remains relatively small for the optimal injection case. Looking at the simulation for the case where the input bunch length is 72 μm, the longitudinal waist is found at 1.911 m from the buncher, 3.1 cm downstream of the zero charge location.

## 4. Bunch Compression Limits for Different Charge Distributions

So far, we assumed a Gaussian temporal profile for the input electron bunch. It is worth investigating the prospect of an ideal phase space such as the uniformly filled ellipsoid with transverse and longitudinal dimensions $a$ and $z_m$, respectively. The line charge density for this distribution is

$$\lambda(z) = \frac{3Q}{4z_m}\left(1 - \frac{z^2}{z_m^2}\right), \tag{32}$$

where $Q$ is the bunch charge, $z_m$ is related to the RMS bunch length of the beam as $z_m = \sqrt{5}\sigma_z$, and likewise $a = \sqrt{5}\sigma_\perp$. Repeating the calculation from Section 2.4.1 of the RF emittance at the exit of the buncher with this line charge density yields

$$\epsilon_{Rf}^2 \approx \sigma_{z0}^2\left[\frac{8}{7}\eta_2^2\alpha^4k^4\sigma_{z0}^4 + \frac{50}{1323}\left(\alpha\eta_1 - 6\alpha^3\eta_3\right)^2k^6\sigma_{z0}^6\right]. \tag{33}$$

This expression is similar to what we previously calculated in Equation (15), but with smaller coefficients (respectively, by 55% and 22%) since, for the same RMS bunch length, the parabolic current profile extends over a smaller interval of RF phases than the long-tailed Gaussian distribution, thus reducing the associated RMS emittance growth.

Reiser offers the perveance and geometry parameters of the longitudinal field evolution [18]. The longitudinal field is given by

$$E_z = -\frac{v(A)}{2\pi\epsilon_0}\frac{\partial\lambda}{\partial z}. \tag{34}$$

Then, the generalized longitudinal force derived from this field yields the perveance

$$K_{L,u} = \frac{g_u(A)Nr_c}{\beta^2\gamma^5}, \tag{35}$$

where now, $g_u(A) = 3v(A)/5\sqrt{5}$. The prefactors of the RF emittance growth and the perveance are smaller than those obtained for Gaussian, so this distribution yields substantial improvements to the optimum bunch length. To this end, we can use the perveance factor and again use the same steps discussed in the previous section to estimate the optimum initial bunch length and the shortest final duration of the beam.

We repeat, for this case, the benchmarking simulation study by initializing in GPT a uniformly filled ellipsoidal distribution with optimal input bunch length and compare the results with the analytical predictions as a function of beam charge as shown in Figure 8a. The agreement is excellent, and it is worthwhile to note that in this case, there is no need to rescale the longitudinal perveance as the space-charge forces are more linear than the Gaussian, so there is less induced emittance growth. Compared to the Gaussian current profile, this distribution improves the optimum bunch length by nearly two.

## 5. X-Band Cavity Compensation

### 5.1. Analytical Estimates

An alternative option to obtain even shorter bunch lengths is using an additional higher-frequency cavity to compensate for the non-linearities imparted on the longitu-

dinal trace space by the buncher RF fields. The use of a higher frequency linearizing cavity has been proposed originally in the context of RF compression by Floettmann [15], and is also very common in FEL beamlines [21] to compensate for the non-linearities in magnetic compressors.

In this section, we consider a setup closely mimicking the current configuration of the UCLA Pegasus beamline. The setup consists of a 1.6 cell S-band (2.856 GHz) gun, an X-band (9.6 GHz) cavity situated at 1.1 m from the cathode plane, and an S-band (2.856 GHz) 11 cell linac. The Linac has a phase chosen to velocity bunch the beam. The linac entrance is 1.4 m downstream of the cathode plane. In Figure 9, we display an illustration of the beamline.

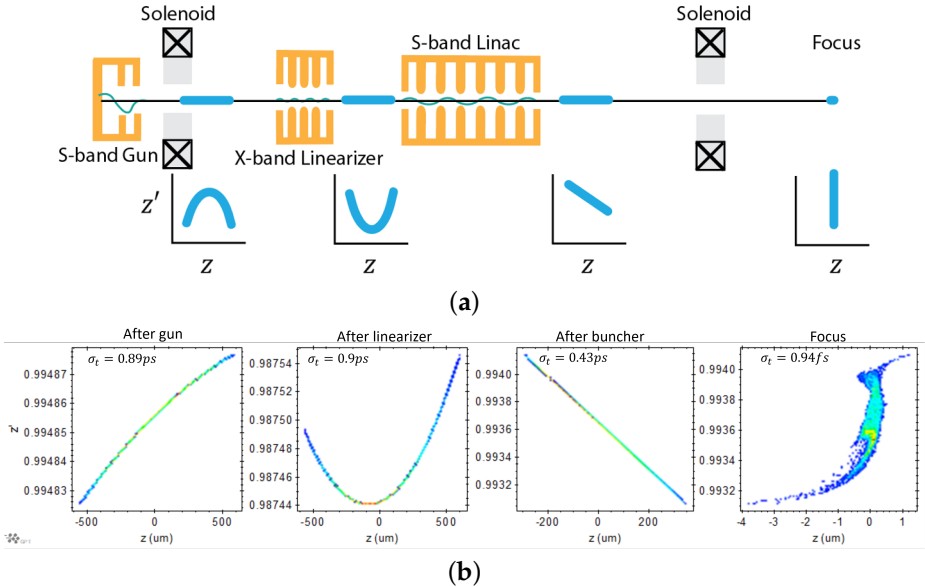

**Figure 9.** (**a**) Illustration of beamline setup for the RF emittance growth compensation. A short x-band cavity is used to compensate for the curvature in $(z, z')$ space imparted by the S-band linac and gun and linearize the output trace space. (**b**) GPT simulation of phase space at the exit of the S-band gun, the exit of the X-band linearizer, the exit of the buncher, and at the longitudinal focus.

An analytical expression for the compensation condition can be found simply imposing the cancellation of the second-order coefficients in the dependence of the relative velocity from the initial longitudinal position yielding

$$\alpha_x = 2\frac{|\eta_2|}{\eta_1}\left(\frac{k}{k_x}\right)^2\alpha^2, \tag{36}$$

where we assume the linearizing cavity to operate at 180 degrees from the crest (i.e., $\Delta\gamma_x = \alpha_x \cos(k_x z_0)$) and $\alpha_x$ and $k_x$ are the normalized voltage and wavenumber of the X-band cavity, respectively. In order to minimize the amount of beam deceleration and the power requirements for the linearizer, it helps to maximize the ratio between the linearizer and buncher cavity frequencies. In addition, for a fixed focal length of the bunching system, $\alpha_x \propto \gamma^5$; thus, lowering beam energy reduces the required voltage. By adding the phase of the linearizer as a degree of freedom, we could simultaneously cancel second- and third-order distortions.

Assuming that the x-band cavity fully compensates the second-order contribution to the emittance and that the cubic RF non-linearity is retained in Equation (27), we can obtain analytical estimates for the minimum achievable bunch length following the same process presented in the previous section. In Figure 10, we plot the optimal solutions for beams of charge 160 fC with Gaussian and uniformly fill ellipsoidal distributions along with the analytical prediction for the uncompensated cases (dashed). The addition of the linearizer cavity allows for achieving bunch lengths significantly shorter (by up to a factor

of five) than the single cavity case. The uncorrelated energy spread sets the thermal limit on the final bunch length even when the space charge is on. In the calculations, we set the uncorrelated energy spread, $\sigma_\delta \approx 10^{-5}$, and for convenience, we show as a dashed line the corresponding thermal limit (0.13 µm).

These results are validated using GPT simulations where we start from an initially uncorrelated coasting beam injected into a beamline with the linearizer cavity and the buncher. A solenoid keeps the aspect ratio between 0.5 and 0.8 as the beam is compressing. The uniformly filled ellipsoidal density analytical prediction matches the simulation results well, while the analytical curve for the Gaussian distribution again underestimates the simulation results due to the non-linear space-charge-induced emittance growth.

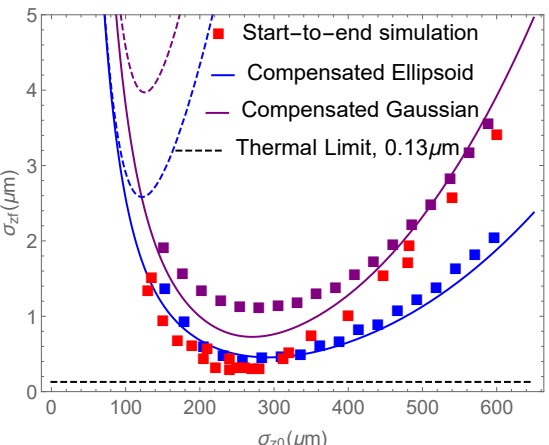

**Figure 10.** Analytical predictions for the RMS bunch length at the longitudinal waist for 160 fC charge and 4.6 MeV energy beams plotted with respect to initial bunch length for compensated (solid) and uncompensated (dashed) cases. Space-charge effects are taken into account assuming a constant aspect ratio equal to 0.6 for Gaussian (purple) and uniformly filled ellipsoidal (blue) distribution. These results are compared with GPT simulations of the linearizer beamline in compensation mode. The red, purple, and blue squares show the results of start-to-end simulations for varying laser pulse lengths following the beam from the cathode located in an S-band RF gun.

In the region along these curves where the space-charge effects are more relevant (i.e., left of the minimum), the final bunch length should increase linearly with charge according to the longitudinal perveance expression from Equation (28). We verify this by simulating an initial bunch length of 180 µm for multiple beam charge between 16 fC and 160 fC. The results are shown in Figure 11 and match well with the analytical predictions. The intercept of the line is due to the finite initial emittance due to thermal and transverse effects [22].

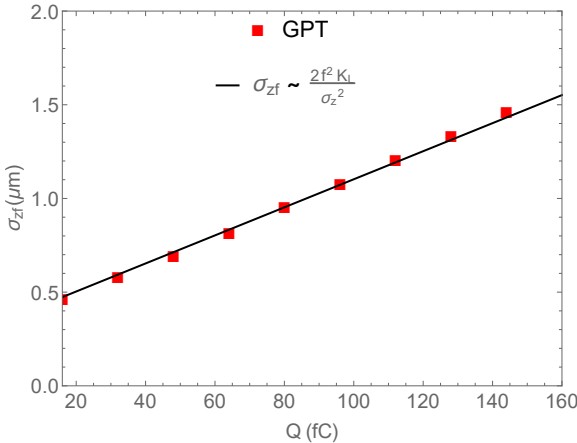

**Figure 11.** Comparison of predicted bunch length dependence on charge for fixed input bunch length. Application to the Gaussian line charge.

### 5.2. Start-to-End Simulations

So far, our discussion has assumed an initially flat and uncorrelated longitudinal phase space. However, in a realistic system, the beam out of the gun would typically present already some correlations (often of non-linear nature) between energy and time along the bunch. To analyze the effects of these, we present start-to-end simulations of the entire beamline, including the RF photo-injector section with nominal parameters listed in Table 2.

**Table 2.** Parameters of start-to-end simluation.

| Parameter | Value |
| --- | --- |
| Charge | $10^6 e$ |
| Laser Spot Size | 10 μm |
| Cathode MTE | 0.5 eV |
| Optimal laser pulse length | 0.95 ps (rms) |
| Gun Accelerating Gradient | 94.7 MV/m |
| Gun Phase | 35.5° |
| Linearizer accelerating voltage | 1.8 MV |
| Linearizer phase | 173.5° |
| Buncher accelerating voltage | 6.75 MV |
| Buncher phase | 101° |
| Final kinetic energy | 4.5 MeV |

The results for minimum waist size as a function of the laser pulse length illuminating the cathode are shown as red squares in Figure 10 and establish a good agreement with the analytical predictions. However, the start-to-end simulations depend on many parameters, such as gun solenoid setting, initial laser spot size, cathode thermal emittance, etc. Consequently, the transverse phase space at the entrance of the RF bunching section was not perfectly matched to the idealized simulations. As a result, the aspect ratio at the focus was 2.8 instead of 1, so the square red dots fall slightly below the analytical estimates, and a minimum bunch length of 0.28 μm or 940 as can be reached.

In Figure 9, we display the simulated phase spaces at various positions along the beamline after each cavity. Figure 12 elucidates the dependence upon the injected bunch length. In the optimal case (center), the initial thermal emittance and the emittance growth imparted by the space charge fields limit the final bunch length. The third-order distortions limit the bunch length if the initial bunch length is too long (right). On the other hand, if the initial injection becomes too short (left), then the final phase space is diluted further by transverse dependence on the longitudinal field, and the focus position significantly moves from the waist plane of the zero space charge case.

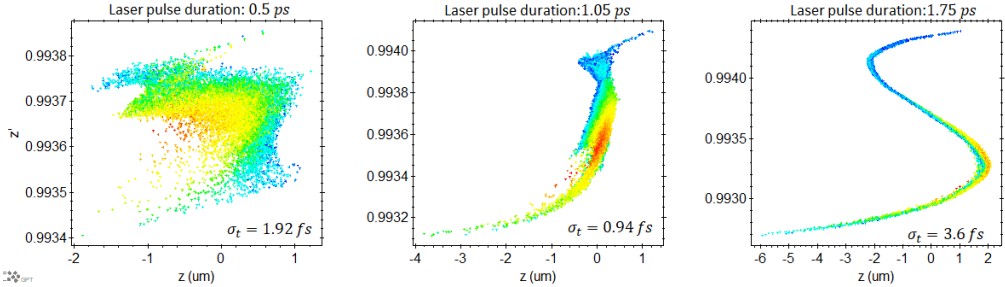

**Figure 12.** Final phase spaces for a beam shorter than, equal to and longer than the optimal bunch length, respectively. The phase spaces are color coded with respect to the radial coordinate.

Finally, we note that while this scheme provides a direct path to sub-fs bunch lengths, the synchronization of the drive signals for two different resonant frequency RF cavities is a significant technological challenge. However, addressing this challenge is the first step before achieving high-quality compensation in a user-facility UED beamline.

Therefore, the case is presented here mainly as an example of the insights offered by the analytical framework that we developed, which suggests that properly shaping the drive laser and compensating the emittance growth does provide a path towards much shorter bunch lengths.

## 6. Conclusions

In conclusion, in this paper, we developed an analytical framework that yields a simple estimate for the minimum bunch length achievable in an RF compression beamline. Besides UED, the formulas in this paper might be helpful in the optimization of RF compression for other ultrashort electron beam applications, including radiation generation and injection into very high-frequency THz-driven [23] and laser-driven accelerators [24].

The envelope formalism allowed us accurate estimation of the final bunch length; our analysis showed that we can integrate the competing non-linear effects into the envelope evolution accurately with relative ease. Specifically, we evaluated longitudinal emittance growth caused by RF curvature and relativistic beam transport, then developed an excellent approximation to the space-charge force in the envelope equation. The results indicate that an optimum initial bunch length condition (which can be satisfied by adjusting the laser pulse length on the cathode, for example) compromises the RF-induced emittance growth and the effects of the longitudinal self-fields. The simplicity of the reported expressions mainly stems from the fact that we approximated the coupling of the transverse and longitudinal space charge dynamics with a simple constant order-of-unity geometry factor in the longitudinal perveance. We also limited the expansion yielding the non-linear terms in the emittance growth to second and third orders. Still, in principle, we could have evaluated all the higher-order terms (and compensated if enough independent knobs were added/available on the beamline).

Although realistic beams produced by photoinjectors typically present more complex phase-space distributions, the initial conditions assumed in the derivations are an initially unchirped longitudinal phase space. Nonetheless, the results obtained here still provide valuable estimates of compression limits in a given configuration, which prove helpful as a starting point for numerical optimizations. In addition, the scaling laws proved capable of guiding parameter choices in the design of new setups. Most importantly, these results highlighted the main contributions to the final bunch length and suggested possible paths to improve the compression further and achieve sub-fs bunch lengths.

**Author Contributions:** Conceptualization and Methodology: P.D. and P.M. Validation: P.D. validated expressions with particle tracking code. Writing—Original Draft Preparation: Both P.D. and P.M. contributed to writing the main text. Visualization: P.D. generated all figures. Supervision: P.M. supervised the development of the paper. Funding Acquisition: P.M. acquired funding. All authors have read and agreed to the published version of the manuscript.

**Funding:** This work was partially funded by the National Science Foundation (NSF) (PHY-1734215); National Science Foundation under the STROBE Science and Technology Center Grant No. DMR-1548924 and the STROBE Consortium MRI grant DMR-1828705.

**Acknowledgments:** The authors want to acknowledge D. Filippetto, A. Kogar and J. Maxson for insightful discussions leading to the calculations in this paper over the years.

**Conflicts of Interest:** The authors declare no conflict of interest.

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
