# Peer review of "Analytical Scaling Laws for Radiofrequency-Based Pulse Compression in Ultrafast Electron Diffraction Beamlines"

_instruments, doi:10.3390/instruments7040049_

Round 1

Reviewer 1 Report

Comments and Suggestions for Authors

This manuscript presents an envelope equation-based approach to obtain analytical scaling laws for the shortest pulse length achievable using radiofrequency based bunch compression.

The paper is  very clear and the described tools are of importance for the design of future photoinejctor based facility. I strongly recommend the paper for pubblication in these Special Issues.  

Author Response

Thank you for accepting.

Reviewer 2 Report

Comments and Suggestions for Authors

In this work, the authors employ the longitudinal envelope equation formalism to highlight the interplay between longitudinal emittance and space-charge forces on the pulse evolution, while keeping the formulas as general as possible for broader applicability to different RF compression setups, whether relativistic or not, and with different RF compression schemes. The authors follow a modular approach, first approximating the buncher with a thin lens, then introducing the space-charge effects to describe different profiles, and -after that- implementing the higher frequency RF cavity that compensates non-linearities. This approach is -in the overall logic, and with all due differences- analogue to https://doi.org/10.1016/j.nima.2012.06.057 - not cited in this work.

The development of analytical frameworks to estimate the achievable bunch length in a beamline through RF compression is relevant across a host of applications, whether at facility-scale or on a tabletop, both for the improvement in machine parameters and for the use of electron bunches to unravel the fastest fundamental physical processes in matter. Either for the implementation of new set-ups, or for the upgrade of pre-existing ones, this work provides the key insights on how laser pulse shaping and emittance growth of the bunches impact bunch lengths, providing a comprehensive, cohesive analysis of all role-playing factors underpinning the achievement of short bunches, and carefully explain -at all steps- the approximations made and their applicability.

For these reasons, and with except to minor typos (see for example) in the "Acknowledgements" section, this work should be accepted for publication.

Author Response

Thanks for accepting the paper and for pointing out the typo in Acknowledgements which has been fixed.

Reviewer 3 Report

Comments and Suggestions for Authors

I have attached my report

Comments on the Quality of English Language

N/A

Author Response

I have attached my reply with corrections.
